# Antiphotoaging Effects of Damiana (*Turnera diffusa*) Leaves Extract via Regulation AP-1 and Nrf2/ARE Signaling Pathways

**DOI:** 10.3390/plants11111486

**Published:** 2022-05-31

**Authors:** Minseon Kim, Lee-Keun Ha, Sarang Oh, Minzhe Fang, Shengdao Zheng, Arce D. Bellere, Jeehaeng Jeong, Tae-Hoo Yi

**Affiliations:** 1Graduate School of Biotechnology, Kyung Hee University, 1732 Deogyeong-daero, Giheung-gu, Yongin-si 17104, Korea; dbs03067@khu.ac.kr (M.K.); vsygn80@khu.ac.kr (L.-K.H.); mincheol1030@khu.ac.kr (M.F.); sdjeong0719@khu.ac.kr (S.Z.); arcedbellere@khu.ac.kr (A.D.B.); 2Snow White Factory Co., Ltd., 807 Nonhyeon-ro, Gangnam-gu, Seoul 06032, Korea; blazma1021@gmail.com (S.O.); dbensk0205@khu.ac.kr (J.J.)

**Keywords:** damiana leaves, skin photoaging, oxidative stress, activator protein-1 (AP-1), nuclear factor erythroid 2–related factor 2 (NRF2)

## Abstract

Damiana (*Turnera di**ffusa*), of the family Passifloraceae, has been widely studied for its pharmacological effects, especially for antioxidant and antibacterial actions. However, there are limited scientific findings describing its antiphotoaging effects on the skin. In the present study, the underlying molecular mechanisms of the protective effect of Damiana were investigated in keratinocytes (HaCaTs) and normal human dermal fibroblasts (HDFs) subject to UVB irradiation. The mRNA expression of matrix metalloproteinases (MMPs) and procollagen type I was determined by reverse transcription-polymerase chain reaction. The protein expression of antiphotoaging-related signaling molecules in the activator protein-1 (AP-1) and nuclear factor erythroid 2-related factor 2 (NRF2)/antioxidant response element (ARE) pathways was assessed by Western blotting. We observed that Damiana blocked the upregulated production of reactive oxygen species induced in UVB-irradiated HaCaTs and HDFs in a dose-dependent manner. Treatment with Damiana also significantly ameliorated the mRNA expression of MMPs and procollagen type I. In addition, the phosphorylation level of c-Jun and c-Fos was also decreased through the attenuated expression of p-38, p-ERK, and p-JNK after treatment with Damiana. Furthermore, the treatment of cells with Damiana resulted in the inhibition of Smad-7 expression in the TGF-β/Smad pathway and upregulated the expression of the Nrf2/ARE signaling pathway. Hence, the synthesis of procollagen type I, a precursor of collagen I, was promoted. Collectively, these results provide us with the novel insight that Damiana is a potential source of antiphotoaging compounds.

## 1. Introduction

Physiological aging is a result of genetic causes; therefore, it is difficult to control. However, photoaging caused by environmental effects such as freckles and skin pigmentation due to ultraviolet rays can be controlled artificially.

Ultraviolet radiation can be classified by wavelength: UVA (315–400 nm), UVB (280–315 nm), and UVC (200–280 nm). In the case of UVB, approximately 90% is absorbed by the ozone layer, whereas the remaining 10% reaches the Earth’s surface; thus, it has a greater impact than UVA [1]. UVB affects cells, producing reactive oxygen species (ROS), which induces oxidative stress and can accelerate skin aging in the human body [2].

Moreover, the direct exposure of keratinous cells to external stimuli, including sunlight, as well as continuous exposure to the external environment causes skin aging and oxidative stress. Such condition damages the extracellular matrix (ECM), collagen, and elastin, which provide cell structural and functional support.

Procollagen type I, the most common protein in the skin, supports the texture between the cells that make up the skin and is a precursor to collagen, a component that makes up the skin and muscles [3]. Elastin is a stretchable protein that is distributed in skin tissues such as collagen which is responsible for the expansion and contraction of skin tissues. In human skin with advanced skin photoaging, tissue disruption, cutting, and dispersion of collagen and elastin are most pronounced [4]. These molecules stimulate the expression of matrix metalloproteinases (MMPs), which disrupt collagen fibers present in the extracellular matrix (ECM) of the dermal skin layer, supporting the migration of macrophages through the dermis to injured sites [5].

UVB, which causes skin photoaging, inhibits the receptors of TGF-β, a major synthetic regulator of procollagen type I. It also affects AP-1, a factor that regulates cell proliferation, differentiation, and apoptosis, which results in an increase in collagenase (MMPs) [3]. In addition, UVB causes excessive intracellular ROS, triggering ROS-sensitive pathways such as AP-1, which increases the expression of MMPs in various types of cells, including HaCaTs and human dermal fibroblasts HDFs. MMPs, which are zinc-dependent endopeptidases, play important roles in the decomposition of the underlying ECM and the expression of inflammatory cytokines. UVB induces the overexpression of MMPs in HaCaTs and HDFs. Furthermore, ROS accelerates the phosphorylation of mitogen-activated protein kinase (MAPK) subunits, which results in the activation of a transcription factor activating protein-1 (AP-1) that initiates the transcription of collagenase MMPs [6]. It was also reported that UV radiation downregulated the expression of the transforming growth factor-beta (TGF-β) receptor, which is involved in the synthesis of procollagen type I, through ROS production. In addition, the expression of the TGF-β1 gene is inhibited by blocking of the Smad signaling pathway by AP-1. Thus, the concentration of ROS should be suppressed to sustain collagen synthesis, which can subsequently maintain the integrity and elasticity of the skin [6].

To defend against the excessive production of ROS, cells can activate the antioxidative system called nuclear erythroid 2-related factor (NRF2). A high level of intracellular ROS frees cytosolic NRF2 from its inhibitor, Kelch-like-ECH-associated protein 1 (KEAP1), allowing it to translocate to the nucleus. NRF2 initiates the transcription of cytoprotective molecules such as NAD (P)H quinone oxidoreductase-1 (NQO1), which removes quinones from biological systems as a detoxification reaction, and heme oxygenase-1 (HO-1), which cleaves oxidative radicals of heme groups [7,8].

*T. diffusa*, known as Damiana, belongs to the Passifloraceae family which is a shrub native [9,10]; it blooms in early to late summer followed by fruits that taste similar to figs. In addition, it is said to have a strong spice-like smell similar to chamomile owing to the essential oils present in it [11]. Furthermore, Damiana leaves are traditionally harvested in the Mexican mountains for tea purposes, herbal smoking mixtures and soaked in sweets [12]. In addition, Damiana leaves contain flavonoids, maltol glucosides, phenols, seven cyanogen glycosides, monophenoids, sesquiterpenoids, triterpenoids, polytephenphenephenol-11, fatty acids, and caffeine, and are specifically separated from *Turnera* flavonoids [13]. The leaves also possess antiviral, antibacterial, antioxidant, and anticancer properties. The pharmacological effects of Damiana leaves have been examined in various studies [14,15]. However, there are few prior studies of the application of Damiana in the treatment of skin aging. Thus, the aim of this study was to demonstrate the recovery effect of Damiana in UVB-irradiated HaCaTs and HDFs and to determine whether Damiana regulates AP-1 and Nrf2/ARE signaling pathways, which are key regulatory biomarkers of skin photoaging.

## 2. Results

### 2.1. Analysis of Chemical Contents of Damiana Leaves Extract

Damiana leaves extract contains a high content of total phenols and flavonoids represented as 101.5 ± 1.67 mg gallic acid/g extract and 108.8 ± 0.01 mg quercetin/g extract, respectively.

As shown in Figure 1, apigenin was identified in Damiana leaves extract at the concentration of 3.66 ± 0.19 mg/g.

### 2.2. Antioxidative Activities of Damiana Leaves Extract

To evaluate the antioxidative activity of the Damiana leaves, its radical scavenging effect was analyzed by DPPH and ABTS assays. As shown in Figure 2, the positive control, ascorbic acid, showed the scavenging effect on DPPH and ABTS radical with an IC_50_ value of 31.71 μg/mL and 82.69 μg/mL, respectively. Damiana leaves extract also significantly suppressed DPPH radicals with an IC_50_ value of 305.4 μg/mL and ABTS radicals with an IC_50_ value of 381.2 μg/mL. These findings reveals that Damiana leaves exhibited a potent antioxidative effect.

### 2.3. Cytotoxicity of Damiana Leaves Extract

The effect of Damiana leaves on cell viability was investigated by MTT assay on HaCaT cells, as shown in Figure 3A. Damiana leaves noted with significant cytotoxicity at 100 µg/mL concentration. Additionally, the effect of Damiana leaves on cell viability was investigated by MTT assay on HDF cells. As shown in Figure 3B, the viabilities of irradiated cells were significantly decreased by 24.6% as compared to non-irradiated cells. Thus, further experiments were conducted on Damiana leaves at lower than 50 µg/mL.

### 2.4. Damiana Leaves Extract Protects HaCaT and HDF Cells from UVB Irradiation

#### 2.4.1. Damiana Leaves Extract on ROS Production in UVB-Irradiated HaCaT and HDF Cells

To measure ROS levels in UVB-irradiated HaCaT and HDF cells were subjected to treatment with the fluorescence dye DCFH-DA. As shown in Figure 4A,B ROS levels were significantly increased by 44.6% and 45.9% in UVB-irradiated cells compared with non-irradiated cells. However, the Damiana leaves treated group showed a significant reduction in ROS compared with the UVB group. Moreover, compared to UVB control cells, treatment of 10 and 50 µg/mL Damiana leaves lowered ROS formation by 27.1% and 36.9%, 51.7% and 65.8%, respectively. It was recorded that Damiana leaves extract was more effective than the positive control ascorbic acid. On the other hand, ascorbic acid inhibited ROS levels by 36% and 50.1%.

#### 2.4.2. Damiana Leaves Extract on the Protein Secretion of MMP-1, MMP-3 in UVB-Irradiated HaCaT, and HDF Cells

Upregulation of MMP-1 and MMP-3 protein levels are associated with collagen degradation. To evaluate the inhibitory properties of Damiana leaves extract to MMP-1 and MMP-3 protein, HaCaT cells were irradiated, and the cell culture supernatant was quantified using an ELISA kit. Findings revealed that UVB irradiation of HaCaT cells increased MMP-1 and MMP-3 protein levels by 27.8% and 8.2% in HaCaT and 49.4% and 48.5% in HDF cells, respectively (Figure 5). Meanwhile, HaCaT cells with Damiana leaves extract downregulated in a dose-dependent MMP-1 and MMP-3 protein production. It was revealed that Damiana leaves extract could inhibit the MMP-1 and MMP-3 protein levels by 48.4% and 30.6% in HaCaT and 49.4% and 48.5% in HDF cells compared to the irradiated control group, respectively.

#### 2.4.3. Damiana Leaves Extract on the mRNA and Protein Expression of MMP-1 and Procollagen Type I in UVB-Irradiated HaCaT and HDF Cells

Similar to ELISA results, in irradiated control cells, the mRNA expression of MMP-1 was elevated by 23.5%, and collagen precursor procollagen type I expression was diminished by 29.3% as compared to the normal group. It was noted that Damiana leaves extract reduced UVB-induced MMP-1 expression by 28.7% (10 μg/mL) and 61.0% (50 μg/mL). Furthermore, Damiana leaves (50 μg/mL) promoted procollagen type I expression by 28.9% (10 μg/mL) and 58.5% (50 μg/mL), respectively, as compared with irradiated control cells. This is incongruent with the positive control ascorbic acid, which showed inhibition by 92.9% on MMP-1 level and upregulation by 43.7% on procollagen type I level, respectively (Figure 6A,B).

The mRNA expression of MMP-1 was elevated by 56.6%; meanwhile, collagen precursor procollagen type I expression was diminished by 76.1%, as compared to the normal group. It was recorded that Damiana leaves extract reduced UVB-induced MMP-1 expression by 74.8% (50 μg/mL). By contrast, Damiana leaves (50 μg/mL) promoted procollagen type I expression by 52.6% as compared with irradiated control cells. This result was comparable to the positive control ascorbic acid, which showed inhibition by 9.6% on MMP-1 level and upregulation by 43.5% on procollagen type I level, respectively (Figure 6C,D).

#### 2.4.4. Damiana Leaves Extract on MMP-1/Procollagen Type Ⅰ Activation in UVB-Irradiated HaCaT and HDF Cells

The cells were treated for 8 h with the indicated concentrations of Damiana leaves and treated with UVB (125 mJ/cm^2^) and UVB (144 mJ/cm^2^). As MMP-1 upregulation is a hallmark of photoaging, protein expression study also indicated an upregulation of MMP-1 by 72.6% which downregulated procollagen type I by 35.7%, compared to normal cells. However, treatment with Damiana leaves at 50 µg/mL reversed this trend, diminished MMP-1 expression by 93.4% as compared to the irradiated control group. Furthermore, Damiana leaves effectively promoted procollagen type I by 61.4% at 50 µg/mL (Figure 7A,B).

As MMP-1 upregulation is a hallmark of photoaging, protein expression study also indicated an upregulation of MMP-1 by 72.9%, consequently downregulating procollagen type I by 25.1%, compared to normal cells. However, treatment with Damiana leaves at 50 µg/mL reversed this trend, diminished MMP-1 expression by 47.3%, compared to the irradiated control group. Furthermore, Damiana leaves effectively promoted procollagen type I by 56.4% at 50 µg/mL (Figure 7C,D).

#### 2.4.5. Damiana Leaves Extract on TGF-β1/Smad7 Activation in UVB-Irradiated HaCaT and HDF Cells

TGF-β1 activation plays an essential role in collagen synthesis. Under UVB (125 mJ/cm^2^) irradiation, TGF-β1 was inhibited by 22.1% due to an increase in inhibitor Smad7 by 35.8%, compared to non-irradiated cells. Damiana leaves recovered the expression level of TGF-β1 by 43.5% (50 μg/mL), while inhibited Smad7 level by 95.5% (50 μg/mL) as compared to UVB-irradiated control group (Figure 8A,B).

Under UVB (144 mJ/cm^2^) irradiation, TGF-β1 was inhibited by 30% due to an increase in inhibitor Smad7 by 21.3%, compared to non-irradiated cells. Damiana leaves recovered the expression level of TGF-β1 by 42.9%, while inhibited Smad7 level by 94.5% at 50 µg/mL, compared to UVB-irradiated control group (Figure 8C,D).

#### 2.4.6. Damiana Leaves Extract on MAPK/AP-1 Activation in UVB-Irradiated HaCaT and HDF Cells

p38, ERK, and JNK are MAPK subunits that can be phosphorylated by UVB irradiation [8]. To validate the Damiana leaves extract role in the mechanism of the MAPKs family, HaCaTs cells were irradiated, and upregulation or downregulation of MAPK subunits was quantified. As shown in Figure 9A,B, UVB triggered an elevation of activated *p*-ERK, *p*-JNK, and *p*-p38. However, treatment with Damiana leaves extracts reversed these changes in a concentration-dependent manner. A supplement of 50 µg/mL Damiana leaves extract suppressed expression of *p*-p38, *p*-ERK, and *p*-JNK by 83.7%, 12.1%, and 45.9%, respectively. Additionally, as shown in Figure 10A,B, UVB irradiation upregulated *p*-c-Fos and *p*-c-Jun. However, treatment of cells with 50 µg/mL Damiana leaves extract inhibited *p*-c-Fos and *p*-c-Jun levels by 25.7% and 42.6%, respectively.

The effect of Damiana leaves extract on MAPKs family members was studied in irradiated HDF cells. As shown in Figure 9C,D, UVB triggered an elevation of activated *p*-ERK, *p*-JNK, and *p*-p38. However, treatment with Damiana leaves extract reversed these changes in a concentration-dependent manner. A supplement of 50 µg/mL Damiana leaves extract suppressed expression of *p*-p38, *p*-ERK, and *p*-JNK by 38.3%, 36.7% and 28.6%, respectively. To further investigate the mechanism of Damiana leaves, we measured protein expression of c-Fos, c-Jun, and their phosphorylated forms. As shown in Figure 10C,D, UVB irradiation upregulated *p*-c-Fos and *p*-c-Jun. However, treatment of cells with 50 µg/mL Damiana leaves extract inhibited *p*-c-Fos and *p*-c-Jun levels by 20.3% and 56.3%, respectively.

#### 2.4.7. Damiana Leaves Extract on Nrf2 Activation in UVB-Irradiated HaCaT and HDF Cells

To assess the antioxidant mechanism of Damiana leaves extract, we evaluated the expression of the antioxidant regulators Nrf2 and Nrf2-related antioxidant proteins in UVB-irradiated HaCaTs. As shown in Figure 11A,B, nuclear Nrf2 protein expression was increased by UVB stimulation. Treatment of cells with 50 µg/mL Damiana leaves extract accelerated the expression of Nrf2 protein by 18.6% compared with UVB irradiation. It was also found that the downregulation of Dihydrolipoamide dehydrogenase (DLD) by 57.8% from the irradiated HaCaT cells was reversed by Damiana leaves treatment by significantly upregulating DLD by 37.5%. Aside from the association of DLD to α-keto acid dehydrogenase, which is a regulator of ROS, it also plays a role in the metabolizing of α-lipoic acid, which activates two cytoprotective proteins, Nrf2 and HO-1. Moreover, HO-1 and NQO-1 levels were dramatically elevated by treatment with Damiana leaves as shown in Figure 11A,B. Furthermore, the expression of HO-1 and NQO-1 protein was increased by 66% and 56.3% with 50 µg/mL Damiana leaves treatment, respectively.

The Nrf2 and Nrf2-related antioxidant proteins in UVB-irradiated HDFs was also explored. As shown in Figure 11C,D, nuclear Nrf2 protein expression was increased by UVB stimulation. Treatment of cells with 50 µg/mL Damiana leaves extract resulted in the acceleration of the expression of Nrf2 protein by 54.3% compared with UVB irradiation. It was elucidated that Damiana leaves extract was able to significantly upregulate DLD by 60.2%. This result is a significant finding given that UVB-irradiated cells downregulated the production of DLD by 16.5%. DLD mechanism leads to the activation of cytoprotective proteins, Nrf2 and HO-1. Moreover, HO-1 and NQO-1 levels were dramatically elevated by treatment with Damiana leaves. As shown in Figure 11C,D, the expression of HO-1 and NQO-1 protein was increased by 58.8% and 32% with 50 µg/mL Damiana leaves treatment, respectively.

## 3. Discussion

Thus, the molecular mechanisms of Damiana on HaCaTs and HDFs cells treated with UVB irradiation in vitro were elucidated. The results showed that Damiana leaves effectively inhibited UVB-induced photoaging by regulating the MMP-1/procollagen type I, TGF-β1/Smad, and MAPK pathways related to skin aging. It was also revealed that Damiana leaves could inhibit the UVB-induced AP-1 mechanism and activate the Nrf2 pathway.

With regard to changes in ROS, ROS formation was decreased by 36.9% on HaCaTs (Figure 4A) and 65.8% on HDFs in the group treated with 50 μg/mL Damiana leaves, which showed a significant 30% effectiveness compared with the positive control (ascorbic acid treatment) (Figure 4B).

The mRNA expression of MMP-1 in HaCaTs was reduced by 61% in the 50 µg/mL treatment group compared with the control group, whereas the mRNA expression of procollagen type I was increased by 58.5%. Ascorbic acid, used as the control treatment, resulted in a 92.9% increase in procollagen type I (see Figure 6A,B). Lin P et al. and Li L et al. claimed that UVB-induced photoaging was reduced by suppressing ROS and MMP-1 [16,17]. In comparison, the results revealed that Damiana possesses potential antiaging properties, mediated through the inhibition of ROS and MMP-1 expression in UVB-irradiated HaCaTs and HDFs.

In addition, the mRNA expression of MMP-1 in HDFs was reduced by 74.8% through treatment with 50 µg/mL Damiana leaves compared with the control group. Furthermore, mRNA expression of procollagen type I was lower in the control group than in the Damiana leaves treatment groups, where a 52.6% increase was noted at 50 µg/mL (Figure 6C,D). These results were comparable to the positive control (ascorbic acid), which resulted in 9.6% inhibition of MMP-1 expression and 43.5% upregulation of procollagen type I expression. Therefore, it can be seen that Damiana leaves exerted a protective effect on collagen expression, by increasing procollagen type I mRNA, which was reduced by UVB which also resulted in a reduction in MMP-1 expression. Wei Gao et al. and Yu Shuai et al. reported that UVB-induced MMP production promotes procollagen type I in HaCaTs and HDFs [18,19]. Damiana leaves were observed to be more efficient in this study; the treatment of HaCaTs with 50 µg/mL Damiana leaves extract increased the protein expression of Nrf2 protein by 18.6% compared with UVB irradiation.

In addition, HO-1 and NQO-1 levels were dramatically elevated by Damiana leaves treatment. As shown in Figure 11A,B, the expression of HO-1 and NQO-1 proteins significantly increased by 66% and 56.3% with treatment at 50 µg/mL. Similarly, in HDFs, treatment of the cells with 50 µg/mL Damiana leaves extract increased the expression of Nrf2 protein by 54.3% compared to UVB irradiation. Furthermore, the protein expression of HO-1 and NQO-1 increased by 58.8% and 32%, respectively, in the 50 µg/mL Damiana leaves treatment group (see Figure 11C,D).

To further investigate the mechanism of Damiana leaves, we measured the protein expression of c-Fos, c-Jun, and their phosphorylated forms and stronger effects on the AP-1 and Nrf2/ARE pathways were found for Damiana treatment than for the positive control. First, in the AP-1 signaling pathway, *p*-c-Fos and *p*-c-Jun decreased by 14.96% and 38.42%, respectively, compared with ascorbic acid with a 50 μg/mL treatment concentration of Damiana leaves in HaCaTs, and decreased by 1.89% and 26.51%, respectively, in HDFs. Second, in the Nrf2/ARE signaling pathway, Nrf2 decreased by 27.13% compared to ascorbic acid at a 50 μg/mL treatment concentration of Damiana leaves, and then HO-1 and NQO-1 also decreased by 38.89% and 17.31%, respectively, in HaCaTs. It was also found that Nrf2 decreased by 29.39% and HO-1 and NQO-1 decreased by 36.57% and 32.28%, respectively, in HDFs. Seo SA et al. and Wang YS et al. reported that UVB-induced damage occurred through the regulation of the AP-1 and Nrf2/ARE pathways in HaCaTs and HDFs [19,20]. Compared with the above studies, it was confirmed that Damiana leaves have excellent efficacy on the AP-1 and Nrf2/ARE mechanism.

In this study, it was confirmed that Damiana leaves extract reduced the AP-1 signaling activity, and upregulated the expression of Nrf2/ARE signaling antioxidant enzymes, which inhibit cell damage from ROS generation, and thereby protect HaCaTs and HDFs from photoaging and cell damage (see Figure 12).

The findings in this study, show that Damiana has significant potential as a biotic molecule that can be utilized in the treatment of photoaging. This study presents the discovery of a novel pharmacological potential of Damiana.

## 4. Materials and Methods

### 4.1. Sample Preparation

A total 100 g Damiana leaves was extracted in 500 mL of 70% ethanol and constantly shaken for 24 h by a Twist shaker at room temperature. The extraction procedure was done thrice. The extracts were collected and subsequently filtered using filter paper (Whatman, Maidstone, Knent, UK). Then, the sample was concentrated by rotary vacuum evaporation (EYELA WORLD–Tokyo Rikakikai Co., LTD., Tokyo, Japan) at 40 °C. The resulted extract yielded 7.38%.

### 4.2. Materials

Damiana leaves powder was purchased from Ecuadorian Rainforest, LLC (Clifton, NJ, USA). Dulbecco’s modified Eagle’s medium (DMEM), fetal bovine serum (FBS), and penicillin-streptomycin were supplied by Gibco RBL (Grand Island, NY, USA). Standard apigenin, positive control ascorbic acid, dexamethasone, and 5-diphenyltetrazolium bromide (MTT) were purchased from Sigma-Aldrich (St. Louis, MO, USA). Human Total MMP-1 and Human Total MMP-3 ELISA kits were purchased from R&D Systems (Minneapolis, MN, USA). Organic solvents were purchased from Samchun Chemical (Seoul, Korea) and Daejung Chemical & Metal (Siheung, Korea). Inorganic salts were purchased from Sigma-Aldrich. Silica gel was purchased from Merck (Kenilworth, NJ, USA). The primary and secondary antibodies were obtained from Cell Signaling Technology (Beverly, MA, USA), Santa Cruz Biotechnology (Santa Cruz, CA, USA), and Bio-Rad Laboratories, Inc. (Hercules, CA, USA).

### 4.3. Total Phenolic and Flavonoid Contents

The total phenolic content of Damiana leaves extract was examined based on Folin–Ciocalteu colorimetric method [5]. Briefly, either gallic acid (6.25–100 µg/mL) standard or plant extract was reacted with 1M Folin–Ciocalteu reagent for 15 min. Then, 0.7 M sodium carbonate in NaOH was added, and the mixture was incubated for 1 h. The absorbance value was measured at a wavelength of 625 nm.

The total flavonoid content of Damiana leaves extract was quantified based on aluminum chloride colorimetric method [6]. First, 50 mg/mL sodium nitrate was mixed with either standard quercetin (0.03125–1 mg/mL) or plant extract. After incubation for 5 min, aluminum chloride was reacted with the mixture for an additional 6 min. Finally, 1M sodium hydroxide was added and incubated for 40 min. Optical density was determined at a wavelength of 450 nm.

The measurement was performed by a microplate reader (Molecular Devices FilterMax F5; San Francisco, CA, USA). The total phenols and flavonoids were presented as gallic acid and quercetin equivalents in mg per gram of plant extract, respectively.

### 4.4. HPLC Analysis

The plant extract was prepared in 50% methanol at the concentration of 2 mg/mL. Serial dilutions (2.5, 25, 125, 250, 500, and 1000 µg/mL) of standard compounds (apigenin) were prepared in methanol. High-performance liquid chromatography (HPLC) was performed on a Dionex Chromelon TM chromatography data system with P580 and UVD100 detectors (Thermo Fisher Scientific Inc., Waltham, MA, USA). Chromatographic separation was performed on a Discovery C_18_ (250 × 4.6 mm, 5-µm particle size). Column temperature was 25 °C; flow rate was 1.0 mL/min; injected volume was 10 µL.

### 4.5. 2,2-Diphenyl-1-Picrylhdrazyl Radical Scavenging Activity

The antioxidant of Damiana leaves extract on 2,2-diphenyl-1-picrylhdrazyl (DPPH, PubChem CID: 2375032) was examined. Various concentrations of Damiana leaves (31.25 –1000 µg/mL) were also tested. Ascorbic acid was used as the positive control [7]. The 0.2 mM DPPH in 100% methanol solution was prepared. An aliquot of 40 µL sample was reacted with 160 µL of DPPH solution, followed by dark incubation at 37 °C for 30 min. The optical density was determined at a wavelength of 595 nm. The inhibitory effect of the sample was assessed using the following Equation:(1)$$\mathrm{DPPH}\;\mathrm{radical}\;\mathrm{inhibition}\;(\%)=\frac{{(\mathrm{OD}}_{0}{-\mathrm{OD}}_{x})}{{\mathrm{OD}}_{0}}\;\times \;100$$

OD_0_: Optical density of negative control; OD_x_: Optical density of the sample.

### 4.6. 2,2′- Azino-Bis (3-Ethylbenzothiazoline-6-Sulfonic Acid) (ABTS) Radical Scavenging Activity

The antioxidant effect of Damiana leaves extract on ABTS (ABTS, PubChem CID: 5464076) was detected. Various concentrations of Damiana leaves (31.25–1000 µg/mL) were also evaluated. Ascorbic acid was used as a positive control. A solution of ABTS was made from the reaction of a 2.5 mM ABTS solution with 1 mM 2,2′-azobis(2-amidinopropane) dihydrochloride (AAPH) and 150 mM sodium chloride. Then, the solution was incubated at 70 °C for 30 min. In each well of a 96-well plate, an aliquot of 4 µL sample was reacted with 196 µL of ABTS solution, followed by dark incubation at 37 °C for 10 min. The optical density was determined at a wavelength of 405 nm. The inhibitory effect of the sample was assessed using the following Equation:(2)$$\mathrm{ABTS}\;\mathrm{radical}\;\mathrm{inhibition}\;(\%)=\frac{{(\mathrm{OD}}_{0}{\;-\;\mathrm{OD}}_{x})}{{\mathrm{OD}}_{0}}\;\times \;100$$

OD_0_: Optical density of negative control; OD_x_: Optical density of the sample.

### 4.7. Cell Culture and Treatment

Murine macrophage Raw264.7 cells were provided by Korean Cell Bank (Seoul, Korea). Human keratinocyte (HaCaTs) by Korean Cell Bank (Seoul, Korea) and Normal Adult Human Primary Dermal Fibroblasts (HDFs) (ATCC PCS-201-012) were purchased from ATCC (Manassas, VA, USA). The cells were grown in an incubator at 37 °C under a humidified atmosphere containing 5% CO_2_. DMEM medium supplemented with 10% heat-inactivated FBS, 1% antibiotics, and antimycotic solution was used for cell culture.

To induce inflammatory responses, Raw264.7 cells were sensitized with 1 µg/mL LPS at the cell confluence of 80% for 24 h. An aliquot of 10 µM dexamethasone (positive control) or Damiana leaves extract (1–50 µg/mL) was diluted in serum-free medium and supplemented at the same time of LPS treatment.

To mimic the photoaging process, after HaCaT and HDF cells reached 80–90% cell confluence, cell plates with closed lids were exposed to UVB (125 or 144 mJ/cm^2^) radiation using UVB irradiation machine (Bio-Link BLX-312; Vilber Lourmat GmbH, Eberhardzell, Germany). Irradiance (0.1 mW/cm^2^) was measured using a UVB photometer (IL1700 Re-search Radiometer/Photometer; International Light, Peabody, MA, USA). Then, cells were rinsed thrice with warm 1X PBS to remove apoptotic cells. Subsequently, fresh serum-free medium containing 10 µM ascorbic acid (positive control) or three doses of Damiana leaves (1–50 µg/mL) were added to each plate for incubation.

### 4.8. MTT Assay

After 24 h of treatment with lipopolysaccharide (LPS) or 72 h of treatment with UVB, 1 mg/mL MTT was added to the cell culture and then incubated for 3 h. After incubation, the medium was discarded, followed by the addition of DMSO to solubilize formazan. The optical density was recorded at a wavelength of 595 nm.

### 4.9. NO Assay

NO production was measured in LPS-induced Raw264.7 cells. Raw264.7 cells were seeded at the density of 1 × 10^6^ cells/mL in 96-well cell culture plates (SPL Life Sciences Co., Ltd., Gyeonggi, Korea) and were incubated for 24 h. Then, 24 h after LPS sensation, the secretion of NO was quantified in the cell culture supernatant. A volume of 100 µL of cell culture supernatant was reacted with 100 µL Griess reagent, a mixture of 1% sulfanilamide in 5% phosphoric acid and 0.1% N-(1-Naphthyl) ethylenediamine dihydrochloride (1:1 ratio). Then, the plate was incubated for 10 min at 37 °C. The absorbance density was measured at 595 nm.

### 4.10. ROS Assay

Intracellular ROS levels were measured in UVB-exposed HaCaT and HDF cells. After 24 h of sample treatment and sensitizer exposure, the supernatant was discarded, and the cells were incubated with 30 µM 2′7′-dichlorofluorescein diacetate (DCFH-DA) (Sigma-Aldrich, St. Louis, MO, USA) for 30 min at 37 °C under the dark condition. Then, the cells were rinsed two times with cooled 1X PBS and collected by using 0.25% and 0.05% trypsin EDTA. Quantitation of intracellular ROS was evaluated by a BD Accuri C6 flow cytometer system (BD Accuri C6, Ann Arbor, MI, USA). The data were collected and analyzed by using FCS 6 plus Research Edition software.

### 4.11. Enzyme-Linked Immunosorbent Assay

HaCaT and HDF cells were seeded at the density of 1.5 × 10^5^ cells/mL in 35 mm cell culture plates to acquire treatment conditions after 24 h. After 72 h of UVB irradiation, cell supernatant was collected and measured [7].

The concentrations of MMP-1 and MMP-3 protein in media were estimated using commercially available ELISA kits following the manufacturers’ instructions. Each sample was repeatedly analyzed twice.

### 4.12. Reverse Transcriptase (RT)-PCR

Cells were collected 24 h after sensitization with inducers. RNA was isolated using TRIZOL reagent following the manufacturer guidelines (Invitrogen Life Technologies, Carlsbad, CA, USA). An equal amount of RNA (3 µg) was reverse transcribed using PCR premix (Bioneer Co., Daejeon, Korea), 0.5 µg/mL oligo-(dT)15 primer, and 0.5 µg/mL hexamer primer. The cDNA was resynthesized at 42 °C for 60 min and was incubated at 94 °C for 5 min to stop the reaction. Amplified products were observed by gel electrophoresis and detected by nucleic acid staining (NobleBio Inc., Gyeonggi, Korea) under UV illumination. GAPDH was used for normalization.

### 4.13. Western Blot

Cell lysates were merged in RIPA buffer (Sigma-Aldrich, St. Louis, MO, USA) for at least 1 h and centrifuged at 12,000 rpm in 15 min to obtain total protein extract. Protein concentration was calibrated using Bradford reagent (Bio-Rad, Hercules, CA, USA). Homogenized proteins were separated by SDS-PAGE and transferred to a nitrocellulose membrane (Bio-rad). Transfer membranes were blocked in 5% skim milk or 5% BSA for 30 min. After several washing steps with 1X TBST, the primary membrane was added with primary antibodies overnight at 4 °C. Subsequently, after incubation with secondary antibody for 1 h, the protein bands were detected using chemiluminescence detection ECL reagents (Fujifilm, LAS-4000, Tokyo, Japan) and ImageMaster™ 17 2D Elite software, version 3.1 (Amersham Pharmacia Biotech, Piscataway, NJ, USA). β-actin and histone were used for normalization of either total protein extract or nuclear protein extract, respectively.

### 4.14. Statistical Analysis

The data were analyzed using Statistical Analysis System (GraphPad Prism 5). All experiments were conducted with three replications. Data are shown as mean ± standard deviation (SD). Significant differences between different treatments were analyzed using a one-way analysis of variance followed by Duncan’s test. The comparison between sample treatments and the control group was performed by using Student’s *t*-tests. Statistical significance was set as follows: * *p* < 0.05, ** *p* < 0.01, and *** *p* < 0.001.

## Figures and Tables

**Figure 1 plants-11-01486-f001:**
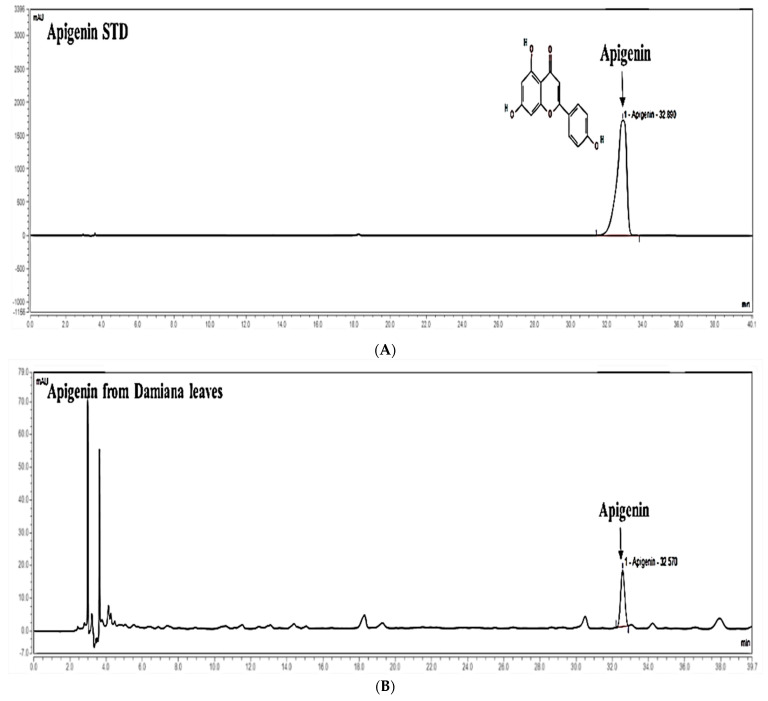
HPLC analysis of standard apigenin (**A**) and Damiana leaves extract (**B**) at 330 nm.

**Figure 2 plants-11-01486-f002:**
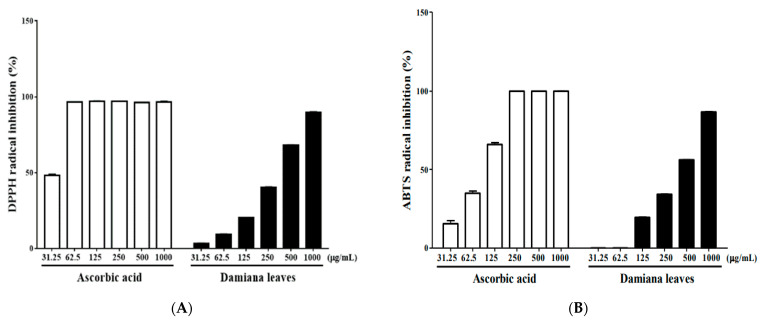
DPPH (**A**) and ABTS (**B**) inhibition of Damiana leaves extract. The results were shown as the mean ± SD of three independent experiments.

**Figure 3 plants-11-01486-f003:**
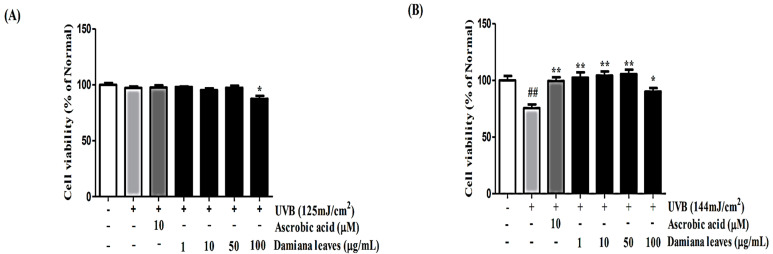
Effect of Damiana leaves extract on cell viability of HaCaT (**A**) and HDF (**B**) cells. Data are presented as the mean ± SD. ^#^ and * indicate significant differences between the non-treated cells and induced groups, respectively. ^##^
*p* < 0.01 vs. the non-treated group. * *p* < 0.05 and ** *p* < 0.01 vs. the induced control.

**Figure 4 plants-11-01486-f004:**
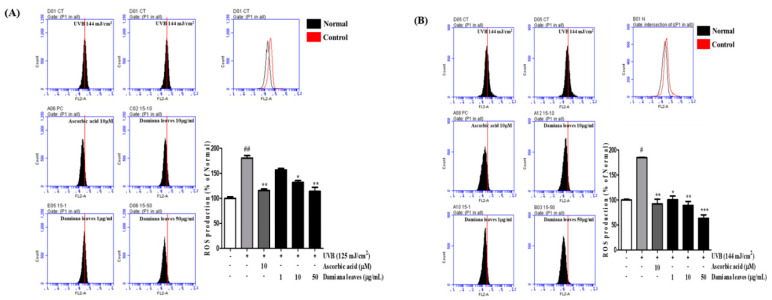
Effect of Damiana leaves extract on levels of intracellular reactive oxygen species (ROS) in UVB-irradiated HaCaT (**A**) and HDF (**B**) cells. After 24 h of treatment, intracellular ROS level was measured. The number of cells is plotted versus the dichlorofluorescein fluorescence detected by the FL-2 channel; results presented as histograms. Data are presented as the mean ± SD. ^#^ and * indicate significant differences from the non-irradiated control and UVB-treated groups, respectively. ^#^
*p* < 0.05 and ^##^
*p* < 0.01 vs. the non-treated group. *, ** and *** *p* < 0.05, 0.01, and 0.001 vs. the UVB-treated control, respectively.

**Figure 5 plants-11-01486-f005:**
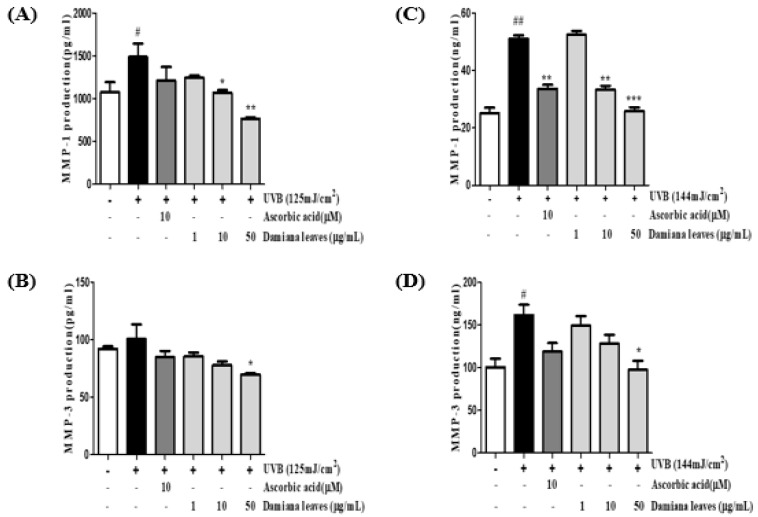
Effect of Damiana leaves extract on the protein secretion of MMP-1 and MMP-3 in UVB-irradiated HaCaT (**A**,**B**) and HDF (**C**,**D**) cells. Data are presented as the mean ± SD. ^#^ and * indicate significant differences from the non-irradiated control and UVB-treated groups, respectively. ^#^ and ^##^
*p* < 0.05 and 0.01 vs. the non-treated group, respectively. *, ** and *** *p* < 0.05, 0.01, and 0.001 vs. the UVB-treated control, respectively.

**Figure 6 plants-11-01486-f006:**
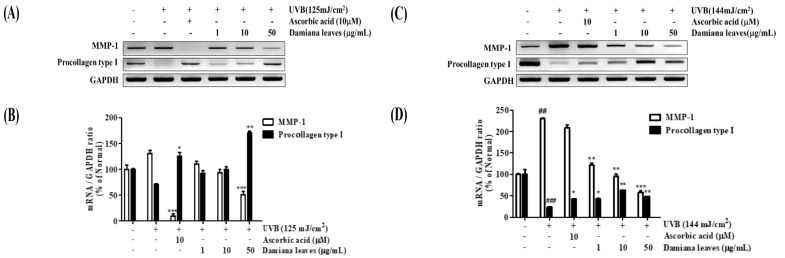
Effect of Damiana leaves extract on mRNA levels MMP-1 and procollagen type I in UVB-irradiated HaCaT (**A**,**B**) and HDF (**C**,**D**) cells. Data are presented as the mean ± SD. ^#^ and * indicate significant differences from the non-irradiated control and UVB-treated groups, respectively. ^##^ and ^###^*p* < 0.01 and 0.001 vs. the non-treated group, respectively. *, ** and *** *p* < 0.05, 0.01, and 0.001 vs. the UVB-treated control, respectively.

**Figure 7 plants-11-01486-f007:**
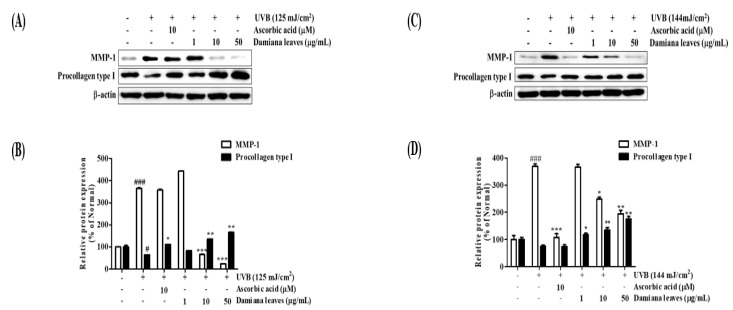
Effect of Damiana leaves extract on protein expression of MMP-1 and procollagen type I in UVB-irradiated HaCaT (**A**,**B**) and HDF (**C**,**D**) cells. Data are presented as the mean ± SD. ^#^ and * indicate significant differences from the non-irradiated control and UVB-treated groups, respectively. ^#^ and ^###^
*p* < 0.05 and 0.001 vs. the non-treated group. *, ** and *** *p* < 0.05, 0.01, and 0.001 vs. the UVB-treated control, respectively.

**Figure 8 plants-11-01486-f008:**
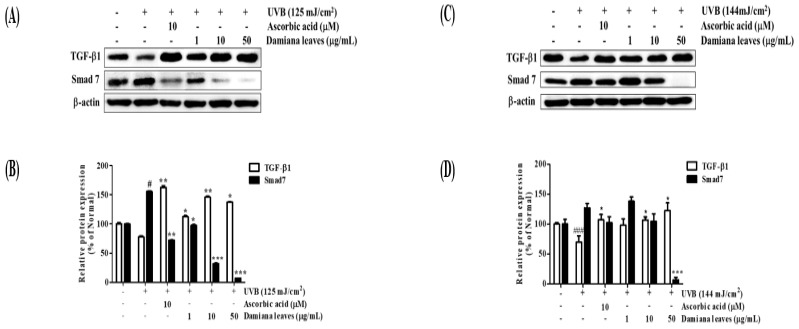
Effect of Damiana leaves extract on protein expression TGF-β1/Smad7 signaling pathway on UVB-irradiated HaCaT (**A**,**B**) and HDF (**C**,**D**) cells. Data are presented as the mean ± SD. ^#^ and * indicate significant differences from the non-irradiated control and UVB-treated groups, respectively. ^#^ and ^###^
*p* < 0.05 and 0.001 vs. the non-treated group. *, ** and *** *p* < 0.05, 0.01, and 0.001 vs. the UVB-treated control, respectively.

**Figure 9 plants-11-01486-f009:**
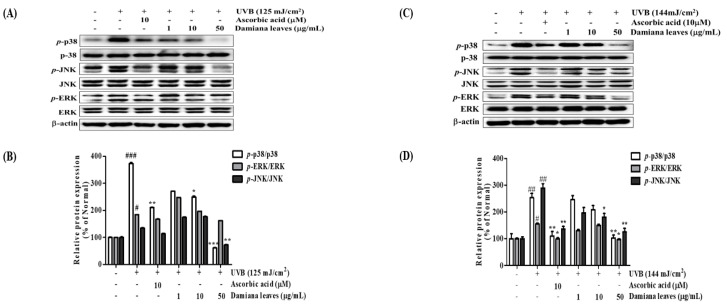
Effect of Damiana leaves extract on protein expression of phosphorylated MAPK signaling pathway on UVB-irradiated HaCaT (**A**,**B**) and HDF (**C**,**D**) cells. Data are presented as the mean ± SD. ^#^ and * indicate significant differences from the non-irradiated control and UVB-treated groups, respectively. ^#^, ^##^ and ^###^*p* < 0.05, 0.01 and 0.001 vs. the non-treated group. *, ** and *** *p* < 0.05, 0.01, and 0.001 vs. the UVB-treated control, respectively.

**Figure 10 plants-11-01486-f010:**
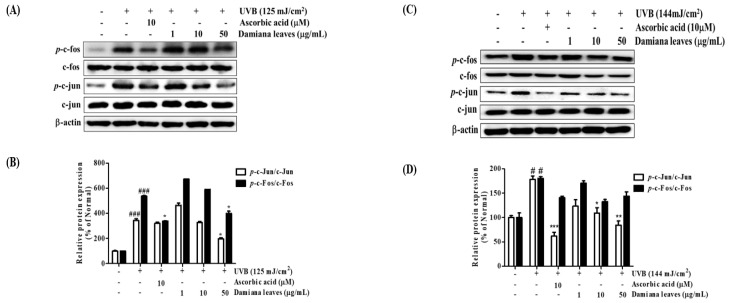
Effect of Damiana leaves extract on protein expression of phosphorylated c-fos and c-jun signaling pathway on UVB-irradiated HaCaT (**A**,**B**) and HDF (**C**,**D**) cells. Data are presented as the mean ± SD. ^#^ and * indicate significant differences from the non-irradiated control and UVB-treated groups, respectively. ^#^ and ^###^
*p* < 0.05 and 0.001 vs. the non-treated group, respectively. *, ** and *** *p* < 0.05, 0.01, and 0.001 vs. the UVB-treated control, respectively.

**Figure 11 plants-11-01486-f011:**
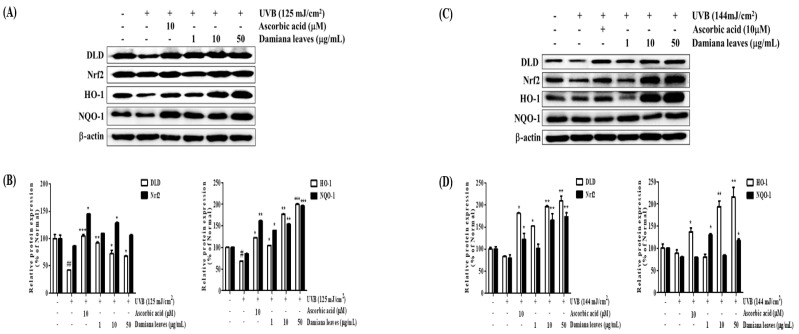
Effect of Damiana leaves extract on protein expression Nrf2, DLD, HO-1, and NQO-1 signaling pathway on UVB-irradiated HaCaT (**A**,**B**) and HDF (**C**,**D**) cells. Data are presented as the mean ± SD. ^#^ and ^*^ indicate significant differences from the non-irradiated control and UVB-treated groups, respectively. ^#^ and ^##^
*p* < 0.05 and 0.01 vs. the non-treated group. *, ** and *** *p* < 0.05, 0.01, and 0.001 vs. the UVB-treated control, respectively.

**Figure 12 plants-11-01486-f012:**
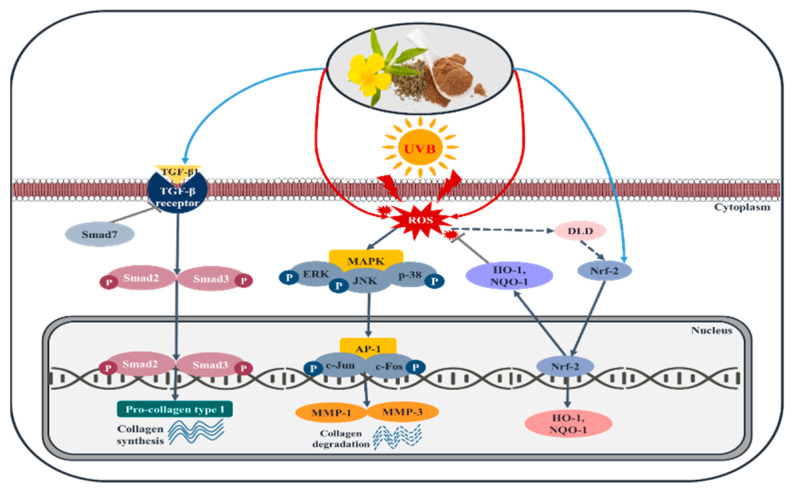
Damiana leaves on skin photoaging signaling.

## Data Availability

The data presented in this study are available in the main text.

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
