# Peer review of "Antiphotoaging Effects of Damiana (Turnera diffusa) Leaves Extract via Regulation AP-1 and Nrf2/ARE Signaling Pathways"

_plants, 2022, doi:10.3390/plants11111486_

Round 1
Reviewer 1 Report
1.Figures 2, 3 and 6 have unreadable written text
2.Figure 4 appears twice numbered, the second must be 5
3. For the HPLC analysis the table of results must be presented, a single chromatogram is not conclusive4.The same type of extract was used for all determinations?
5. In point 4.4. the Folin method and total flavonoids are described, but the result is not presented anywhere
6. In point 4.4. the equation is unclear
Author Response
May 24nd, 2022
Plants
Dear Editor-in-Chief,
It is my pleasure to submit a manuscript titled, “Anti-photoaging Effects of Damiana (Turnera diffusa) Leaves Extract via Regulation AP-1 and Nrf2/ARE Signaling Pathways,” which I wish to be considered for publication in plants.
Turnera diffusa (also known as Damiana) has been traditionally and scientifically studied for the treatment of antiviral, antibacterial, antioxidant, and anticancer; however, the publication about the application of Damiana on skin diseases is limited, especially in skin photoaging. In the current study, ethanolic extract from Damiana was investigated in vitro for its biological effects on UVB-irradiated keratinocytes and dermal fibroblasts.
First, I changed the figure part that the reviewer pointed out and put it back in. Also, I would like to add the supplementary information about the part you told me to put in the table for HPLC analysis, is that okay?
Thank you for reviewing my paper.
Sincerely,
Tae-Hoo Yi, K.M.D, Ph.D.
College of Life Science, Kyung Hee University. 1732 Deogyeong-daero, Giheung-gu, Yongin-si, Gyeonggi-do 17104, Korea. E-mail: drhoo@khu.ac.kr

Reviewer 2 Report
Opinion related to the paper entitled “Anti-photoaging effects of Damiana (Turnera diffusa) leaves extract via regulation AP-1 and Nrf2/ARE signalling pathways”.
Most of the UVB radiation is absorbed by the ozone layer but at low intensity, reaches earth’s surface. Due to the high biological activity, it strongly influences the biochemical parameters of the skin, accelerating the ageing process and in the case of excessive exposure, it may lead to the development of cancer.
Authors used keratinocytes or dermal human fibroblasts as a model. They clearly showed that the extract from the leaves Turnera diffusa activates positive processes for the skin (procollagen type I level, TGF-β1 activation) and inhibits harmful processes (Smad7 level, ROS level, MMP-1 production).
The work is worth publishing after minor corrections.
The strengths of this work is the use of advanced techniques, unambiguous demonstration of beneficial biological activity of extract, clear presentation of the results.
The weakness is the poor, preliminary phytochemical analysis of the extract.
Detailed comments.
1. Line 42 UVA 315-400, UVB 280-315, UVC 200-280.
2. Line 62 – better write “collagenase (MMPs)”.
3. Line 85 – should be United States – south part.
4. Line 112- should be in Fig. 2. I am surprised that Figure 2 A and B are identical. I have used ABTS and DPPH many times and the results have always differed. The tendencies were maintained, of course, but I never got such an identity. Maybe there was a mistake?
5. Line 173, 179 – why procollagen type I is written with a capital letter (correct throughout the text).
Author Response
May 24nd, 2022
Plants
Dear Editor-in-Chief,
It is my pleasure to submit a manuscript titled, “Anti-photoaging Effects of Damiana (Turnera diffusa) Leaves Extract via Regulation AP-1 and Nrf2/ARE Signaling Pathways,” which I wish to be considered for publication in plants.
Turnera diffusa (also known as Damiana) has been traditionally and scientifically studied for the treatment of antiviral, antibacterial, antioxidant, and anticancer; however, the publication about the application of Damiana on skin diseases is limited, especially in skin photoaging. In the current study, ethanolic extract from Damiana was investigated in vitro for its biological effects on UVB-irradiated keratinocytes and dermal fibroblasts.
Thank you for pointing it out first. I modified the part you wanted to modify. Also, I revised the data and put it back in, so please check it
Thank you for reviewing my paper.
Sincerely,
Tae-Hoo Yi, K.M.D, Ph.D.
College of Life Science, Kyung Hee University. 1732 Deogyeong-daero, Giheung-gu, Yongin-si, Gyeonggi-do 17104, Korea. E-mail: drhoo@khu.ac.kr
